# NEW RESULTS FOR OPERATOR MICHELSON CONTRAST

## ABSTRACT

We consider the generalization of the Michelson contrast for positive operators of countably decomposable $W^*$-algebras and prove its properties. In addition, we study how the inequalities characterizing traces interplay with the Michelson contrasts of operator variables.

Also, we developed a torch code for simulation modeling code to Monte-Carlo type I von Neumann algebras.

Let $\mathcal{A}$ denote some Banach $*$-algebras; then, $\mathcal{A}^{\mathrm{sa}}, \mathcal{A}^+$ are its self-adjoint and positive parts, respectively. $\mathcal{A}^*$ is the conjugate space of continuous linear functionals. If $\mathcal{A}$ is von Neumann algebra, then $\mathcal{A}_*$ denotes its predual space. Additionally, $\mathcal{A}_*^+$, $\mathcal{A}^{*+}$ are the positive cones in $\mathcal{A}_*$ and $\mathcal{A}^*$, respectively. Tr denotes the canonical trace of $\mathbb{M}_n(\mathbb{C})$. By $C(H)$ and $B(H)$ we denote the ideal of compact operators and the algebra of bounded operators, respectively.

## 1 PRELIMINARIES

From the work (16) we know.

Let $\mathcal{A} = B(H)$, then the center $\mathfrak{C}(B(H))$ of $B(H)$ is equal to $\mathbb{C}\mathbf{1}$. Let us consider the function

$$\Delta(x) = \inf_{A \in \mathbb{R}^+} \left\{ \left\| \mathbf{1} - \frac{x}{A} \right\| \right\} \text{ for } x \in B(H)^+,$$

which illustrates how far the element $x$ is from the central elements. If $x = \mathbf{1}$, then $\Delta(\mathbf{1}) = 0$ ($A = 1$) and $\Delta(\mathbf{0}) = 1$.

**Proposition 1** *Let $x$ be positive operator ($x \in B(H)^+$), then $\Delta(x) \leq 1$.*

**Proposition 2** *Let $x$ be positive non-invertible (singular) operator, then*

$$\Delta(x) = 1.$$

**Corollary 1** *Let $x$ be positive compact operator, then $\Delta(x) = 1$.*

**Theorem 1** *Let $x$ be invertible positive operator ($x \in B(H)^+$), with the inverse $x^{-1}$, then*

$$\Delta(x) = \frac{\|x\|\|x^{-1}\| - 1}{\|x\|\|x^{-1}\| + 1}. \tag{1}$$

**Corollary 2** *Let $x \in B(H)^+$ be invertible element with the inverse element $x^{-1} \in B(H)$, then $\Delta(x) < 1$.*

**Corollary 3** *Let the sequence $x_n$ from $B(H)^+$ that converges to element $\mathbf{0} \neq x \in B(H)^+$ in terms of norm,*

$$\lim_n \Delta(x_n) = \Delta(x),$$

*i.e. $\Delta : (B(H)^+ \setminus \{\mathbf{0}\}, \|\cdot\|) \mapsto [0, 1]$ is a continuous function.*

**Corollary 4** *If the sequence of operators is converging to a non-singular (invertible) operator, then the sequence contains not more than a finite quantity of non-invertible operators.*

**Corollary 5** *For any $x$ in $B(H)^+$ the following properties*

*1.*

$$\Delta(x) = \frac{\sup(\sigma(x)) - \inf(\sigma(x))}{\sup(\sigma(x)) + \inf(\sigma(x))};$$

*2.*

$$\Delta(x) = \frac{\sup \sigma(x)}{\sup \sigma(x) + \inf \sigma(x)} \left\| 1 - \frac{x}{\|x\|} \right\|.$$

*hold.*

The first equality states that the $\Delta$ is indeed the Michelson contrast.

**Theorem 2** *Let $x, y \in B(H)^+$, then $\Delta(x + y) \leq \max\{\Delta(x), \Delta(y)\}$.*

## 2 COMPUTATIONAL EXPERIMENTS

We conducted additional computational experiments on the inequalities violations for the higher dimensions compared to (16).

### 2.1 GARDNER'S INEQUALITY INSPIRED SIMULATION

In 1979 L.T. Gardner showed the inequality $|\varphi(X)| \leq \varphi(|X|)$ characterizes traces in $C^*$-algebras among all functionals, i.e.

**Theorem 3 ((6), Theorem 1)** *The finite traces on a $C^*$-algebra $\mathcal{A}$ are precisely those (positive) linear functionals $\varphi$ on $\mathcal{A}$ which satisfy $|\varphi(x)| \leq \varphi(|x|)$ for all $x \in \mathcal{A}$.*

If $\varphi$ is a tracial functional on the $C^*$-algebra $\mathcal{A}$, then the Gardner exponent shows the result for all elements of $X \in \mathcal{A}$ and, conversely, if for all $X \in \mathcal{A}$ is a Gardner quality indicator and $\varphi$ is a positive functional, this functional is tracial.

Let $\mathcal{M}$ be a von Neumann algebra, the normal strongly semifinite weight $\varphi$ ensures that for any $\varphi$-finite projections $P \in \mathcal{M}$, the Gardner equivalent ( $|\varphi(X)| \leq \varphi(|X|)$) result for all $X = P X_0 P$, where $X_0 \in \mathcal{M}$, then the weight is a trace.

In case if $\mathcal{A} = \mathbb{M}_n(\mathbb{C})$, we have that for $\varphi := \mathrm{Tr}(A \cdot)$ the inequality must be violated for some $X \in \mathbb{M}_n(\mathbb{C})$, i.r. exists $X \in \mathbb{M}(\mathbb{R})$ such that $|\mathrm{Tr}(AX)| - \mathrm{Tr}(A|X|) > 0$.

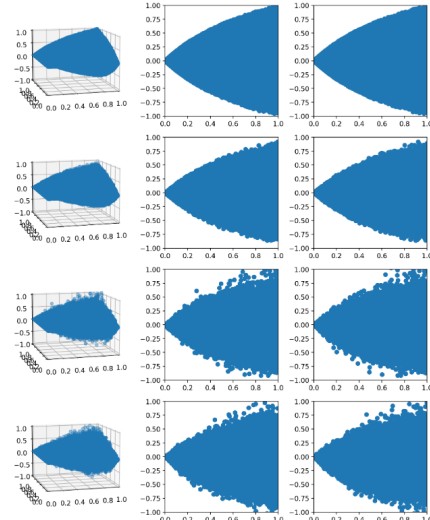

Figure 1: The scatter plots above are visualising results of simulations with $x = \Delta(X)$, $y = \Delta(Y)$, with $X \in \mathbb{M}_n(\mathbb{R})$, $Y \in \mathbb{M}_n(\mathbb{R})$, $\|X\| = \|Y\| = 1$ and $z = |\mathrm{Tr}(XY)| - \mathrm{Tr}(|X||Y|)$. The left column is a 3D scatter plot, the middle column is a plot of $z$ vs. $x$ and the right column is $z$ vs. $y$. The rows correspond for 2, 3, 4 and 5-dimensional simulations respectively.

## 2.2 QUANTUM JENSEN-SHANNON DIVERGENCE

Let $X$, $Y \in \mathbb{M}_n^+(\mathbb{R})$ and $\mathrm{Tr}(X) = \mathrm{Tr}(Y) = 1$.

We call $S(X) := -\mathrm{Tr}(X \log_2(X))$ the von Neumann entropy, where $\log_2(X)$ is understood in the terms of functional calculus.

We define

$$QJSD(X,Y) := S\left(\frac{1}{2}(X+Y)\right) - \frac{1}{2}\left(S(X) + S(Y)\right)$$

following the [(29), (30),(31)].

In the following computational experiment we compare the Michelson Contrast $\Delta(XY)$ of the product of density matrices $X$ and $Y$ with the $\sqrt{QJSD(X,Y)}$.

We see the tendency that if we increase the dimension it seems that the following type inequalities

$$A \times \Delta(|XY|) \leq \sqrt{\mathrm{QJSD}(X,Y)} \leq B \times \Delta(|XY|)$$

occur. It seems logical since we know (26) where authors state the equivalence between Jensen–Shannon divergence and Michelson contrast for a continuous commutative distributions.

## 2.3 $L_1$ EQUALITY VIOLATION

From (18) we now that if $A \in \mathbb{M}_n^+(\mathbb{R})$ and $\mathrm{Tr}(|AXA|) = \mathrm{Tr}(A|X|A)$ for all $X \in \mathbb{M}_n^{sa}(\mathbb{R})$ then $A$ is central.

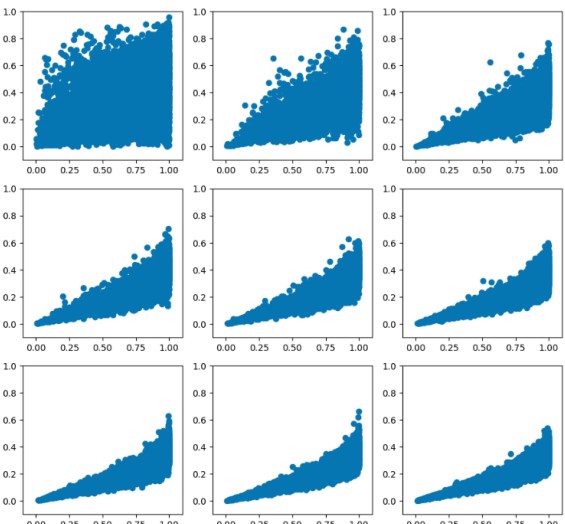

Figure 2: The scatter plots above are visualising results of simulations with $x = \Delta(|XY|)$, $y = \sqrt{QJDS(X,Y)}$, with $X$, $Y \in \mathbb{M}_n^+(\mathbb{R})$, $\mathrm{Tr}X = \mathrm{Tr}Y = 1$. The upper row corresponds to 2, 3 and 4-dimensional simulations, the middle row corresponds to 5, 6 and 7-dimensional simulations and the last row corresponds to 8, 9 and 10-dimensional simulations.

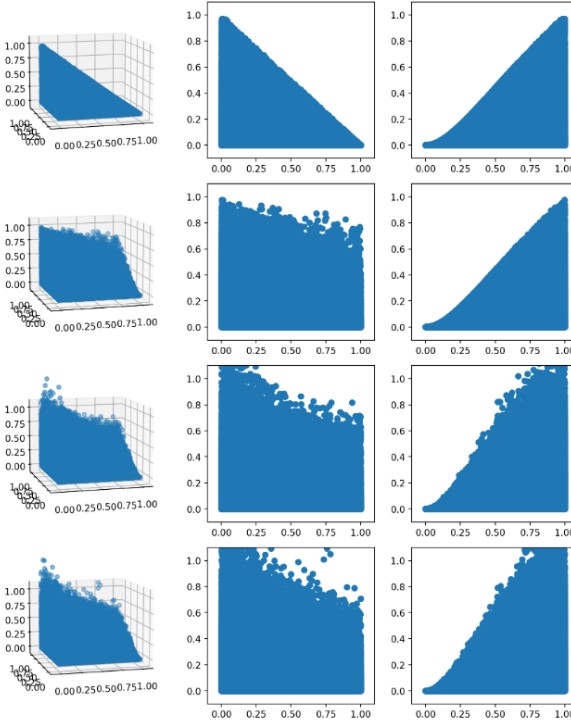

Figure 3: The scatter plots above are visualising results of simulations with $x = \Delta(|X|)$, $y = \Delta(Y)$, with $X \in \mathbb{M}_n^{sa}(\mathbb{R})$, $Y \in \mathbb{M}_n^+(\mathbb{R})$, $\|X\| = \|Y\| = 1$ and $z = \mathrm{Tr}(Y|X|Y)| - \mathrm{Tr}(|YXY|)$. The left column is a 3D scatter plot, the middle column is a plot of $z$ vs. $x$ and the right column is $z$ vs. $y$. The rows correspond for 2, 3, 4 and 5-dimensional simulations respectively.

## 3    LIMIT SIMULATION

The purely new results are obtained on the limits of sums of positive operators.

Let $X_1, \ldots X_n \in \mathbb{M}_n^+(\mathbb{R})$. Consider a sequence $\Delta\left(\frac{1}{n}\sum_{k=1}^{n} X_k\right)$. We know, that $\Delta\left(\frac{1}{n}\sum_{k=1}^{n} X_k\right) = \Delta\left(\sum_{k=1}^{n} X_k\right)$.

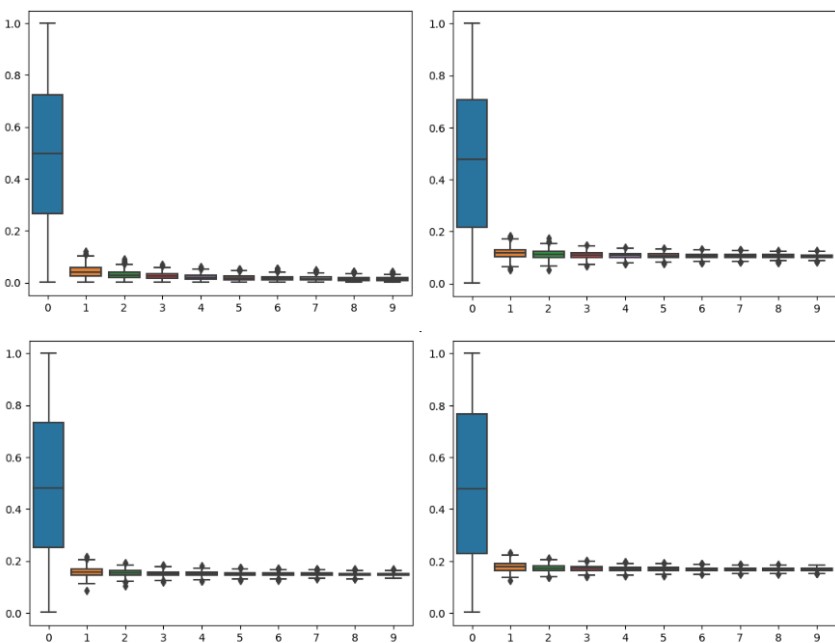

Figure 4: The upper left plot corresponds to 2-dimensional case, upper right to 3-dim, lefter bottom is 4-dim and righter bottom 5-dim.

We see that for the higher dimensions – the higher is the limit of the Michelson contrast of its sum. Yet, in any version it seems to be converging.

## ACKNOWLEDGMENTS

This work is supported by Mathematical Center in Akademgorodok under agreement No.075-15-2022-281 (05.04.2022) with the Ministry of Science and Higher Education of the Russian Federation.

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
