# OpenReview forum: "New results for operator Michelson contrast"
_mathai.club/MathAI/2025/Conference — MathAI 2025 Oral_

### Official Review · Reviewer_JRjK · 2025-02-26
**NEW RESULTS FOR OPERATOR MICHELSON CONTRAST**

**Rating:** 6
**Confidence:** 3

**Review:**

The topic of this paper is related to the so-called Michelson contrast (also called visibility measure), that is widely applied in image processing, optics, and vision, so the talk may be interesting to some participants of MathAI conference.

Most of the contents of this paper are covered by the recent publication:

Abed, S.A., Nikolaeva, I.A. & Novikov, A.A. Generalisation of Michelson Contrast for Operators and Its Properties. Lobachevskii J Math 45, 3835–3848 (2024). https://doi.org/10.1134/S1995080224603400

The only unpublished part seems to be the Figure 4 on page 6.

If the talk is authored by the same researchers as the above mentioned paper, then I suggest that the paper should be included into the conference program, but should not be published because most of the contents have been published elsewhere.

In the unlikely case that the talk has different authors than the paper mentioned above (the authors information is unavailable to me as a reviewer in double-blind reviewing process), the paper should be rejected due to plagiarism.

---

### Official Review · Reviewer_M8Ww · 2025-02-26
**Good paper**

**Rating:** 8
**Confidence:** 4

**Review:**

Strengths:
The paper introduces continuous (everywhere except 0) characteristic delta of operators in $B(H)^+$ which shows centrality of an operator. Paper provides connection of delta of an operator to the its invertability and connection to its norm and norm of its inverse, in cases where inverse exists. Paper also shows connection of delta of operator to the its spectrum.

Authors propose two conjectures. First hypothesis is that connection of delta of composition of operators is equivalent to quantum Jensen-Shannon divergence of two operators. This conjecture is based on similar result for continuous commutative distribution.
Second conjecture is that delta of sum of operators converge if amount of operators approaches infinity.
Both conjectures were tested numerically on large sample of random matrices.

Weaknesses:
It may be beneficial to show why second conjecture (which is "$\Delta\left(y_n\right) = \Delta\left(\sum\limits_{i = 1}^n x_i\right)$ is converging") does not follow from low of large numbers and continuity of delta "$\lim\limits_{n\to\infty} \Delta(y_n) = \lim\limits_{n\to\infty} \Delta\left(\sum\limits_{i = 1}^n x_i\right) = \lim\limits_{n\to\infty} \Delta\left(\frac{1}{n}\sum\limits_{i = 1}^n x_i\right) = \Delta\left(\lim\limits_{n\to\infty} \frac{1}{n}\sum\limits_{i = 1}^n x_i\right)$"

---

### Official Review · Reviewer_ZbCX · 2025-02-27
**New results for operator Michelson contrast**

**Rating:** 8
**Confidence:** 4

**Review:**

The article explores the extension of Michelson contrast to positive operators, providing new theoretical results and computational experiments. The presentation of relevant definitions and theorems is clear and easy to understand, and the theoretical results are validated through computational experiments, supported by a wealth of persuasive graphs and charts. The article is well-structured and logically rigorous.
Regarding the new results on the limits of the sum of positive operators, it would be beneficial to include more analysis of the experimental results, such as comparisons with existing literature or applications of the conclusions.

---

### Decision · Program_Chairs · 2025-03-08

**Decision:**

Accept (Oral)

**Comment:**

Your article has been accepted and you can give a talk on the article. All articles will be sorted by rating and within the available conference places one author from each article will be invited. If there are not enough places, then you will either have the opportunity to speak remotely or come at your own expense!